

# Antimicrobial and anti-inflammatory activity of Cystatin C on human gingival fibroblast incubated with *Porphyromonas gingivalis*

Blanca Esther Blancas-Luciano[1], Ingeborg Becker-Fauser[2], Jaime Zamora-Chimal[2], José Delgado-Domínguez[2], Adriana Ruíz-Remigio[2], Elba Rosa Leyva-Huerta[3], Javier Portilla-Robertson[3] and Ana María Fernández-Presas[1,4]

[1] Departamento de Microbiología y Parasitología, Universidad Nacional Autónoma de México, Mexico City, México
[2] Unidad de Investigación en Medicina Experimental, Universidad Nacional Autónoma de México, Mexico City, México
[3] Departmento de Medicina Oral y Patología, División de Posgrado, Facultad de Odontología, Universidad Nacional Autónoma de México, Mexico City, México
[4] Centro de investigación en Ciencias de la Salud (CICSA), Universidad Anáhuac México Campus Norte, Mexico City, México

Corresponding author
Ana María Fernández-Presas, presas@unam.mx

## ABSTRACT

**Background**. Periodontal disease is considered one of the most prevalent chronic infectious diseases, often leading to the disruption of tooth-supporting tissues, including alveolar bone, causing tooth mobility and loss. *Porphyromonas gingivalis* is considered the major etiological agent of this disease, having a plethora of virulence factors, including, lipopolysaccharides (LPS), hemolysins, and proteinases. Antimicrobial peptides are one of the main components of the innate immune response that inhibit the growth of *P. gingivalis*. The aim of this study was to analyze the antimicrobial activity of cystatin C and to assess the effect on the inflammatory and anti-inflammatory cytokines, the production of reactive oxygen species, and in the release of nitric oxide by human gingival fibroblasts incubated with *P. gingivalis* in the presence and absence of cystatin C.

**Methods**. *P. gingivalis* ATCC 33277 was exposed to cystatin C for 24h and co-cultured with human gingival fibroblasts (HGFs) ATCC CRL-2014. The effect of cystatin on growth of *P. gingivalis* and HGFs was evaluated. Pro-inflammatory (TNF$\alpha$, IL-1$\beta$) and anti-inflammatory (IL-10) cytokines were determined by ELISA in the supernatants of HGFs incubated with *P. gingivalis* exposed to cystatin C. Additionally, nitrites and reactive oxygen species (ROS) production were evaluated.

**Results**. Cystatin C inhibited the growth of *P. gingivalis* without affecting HGFs. Incubation of HGFs with *P. gingivalis* led to a significant increase of TNF-$\alpha$ and IL-1$\beta$. In contrast, HGFs incubated with *P. gingivalis* exposed to cystatin C showed a decreased production of both cytokines, whereas IL-10 was enhanced. Incubation of HGFs with *P. gingivalis* led to an increase of nitric oxide (NO) and ROS production, which was reduced in the presence of the peptide.

**Conclusions**. Cystatin C inhibits the growth of P. gingivalis and decreases the inflammatory cytokines, ROS, and NO production during infection of HGFs with

*P. gingivalis.* Knowledge on the antimicrobial and immunomodulatory properties of cystatin C could aid in the design of new therapeutic approaches to facilitate the elimination of this bacterium to improve the treatment of periodontal disease.

## INTRODUCTION

Periodontitis is a chronic infectious disease, characterized by an exacerbated inflammatory response and progressive loss of tooth supporting tissues (*Könönen, Gursoy & Gursoy, 2019*) *Porphyromonas gingivalis* is a periodontopathogen bacterium implicated as a major, etiological agent in periodontitis (*Van Winkelhoff et al., 2002*). This bacterium has been recovered from periodontal pockets in a high percentage (75.8%) of patients with periodontitis (*Rafiei et al., 2017*).

The most abundant cell types in periodontal connective tissues are gingival fibroblasts (GF), where they participate in the repair of periodontal tissues during inflammatory periodontal diseases (*Lee, Lee & Jang, 2013*). GF also promotes periodontal wound healing (*Smith et al., 2019*; *Baek, Choi & Ji, 2013*).

Furthermore, LPS of *Porphyromonas gingivalis* increases their superoxide concentrations after the exposure to human gingival fibroblasts (HGFs) (*Staudte et al., 2010*; *Gölz et al., 2014*). Thus, these cells can also participate in the progression of periodontitis, inducing the release of inflammatory such as mediators nitric oxide cytokines, and reactive oxygen species (ROS), and nitric oxide (*How, Song & Chan, 2016*; *Kirkwood et al., 2007*; *Gölz et al., 2014*; *Herath et al., 2016*).

Cytokines are involved in the initiation and progression of periodontal disease (*Ramadan et al., 2020*) Even though secreted cytokines promote the elimination of bacteria, the overproduction of pro-inflammatory cytokines may participate directly in periodontal breakdowns, such as the breakdown of collagen periodontal attachment loss, and alveolar bone resorption (*Gabay, Lamacchia & Palmer, 2010*). TNF-$\alpha$ and IL-1$\beta$ are the major secreted pro-inflammatory cytokines, that are important markers of periodontitis progression and severity. They are also the main inducers of effector molecules that cause the breakdown of periodontal tissues (*Gomes et al., 2016*). TNF-$\alpha$ and IL-1$\beta$ are produced by several cell types including dendritic cells, macrophages, periodontal ligament cells, osteoblasts, and gingival fibroblasts and can act as multifunctional molecules (*Cheng et al., 2020*). IL-1 $\beta$ promotes production of metalloproteinases (MMPs), which are involved in the extracellular matrix degradation and, in turn, bone resorption and periodontal tissue destruction (*Aleksandrowicz et al., 2021*). TNF-$\alpha$, participates in the bone resorption process, inducing RANK expression in osteoclast precursors and RANKL expression in osteoblast (*Pan, Wang & Chen, 2019*). In addition, TNF-$\alpha$ and IL-1 $\beta$ also induce reactive oxygen species (ROS) generation in periodontal tissue (*Wang et al., 2014*), where oxidative stress has been shown to be involved in periodontitis (*Tomofuji et al., 2006*; *Maruyama et*

These pro-inflammatory mediators are required for the immune defense against bacteria, yet their uncontrolled activity leads the accumulation of ROS (superoxide radicals, hydrogen peroxide, hydroxyl radicals and singlet oxygen) (*Gölz et al., 2014*). Even though these products stimulate proliferation and differentiation of cultured human periodontal ligament fibroblasts at low concentrations, their presence in higher concentrations can induce pathogen killing and cytotoxic effects on periodontal tissues and pathogen killing (*Chapple & Matthews, 2007*). *Zhu et al. (2020)* demonstrated that after the stimulation *of HGFs* with *LPS*, ROS production in mitochondria (mtROS) was significantly enhanced, these results indicate that oxidative stress can be induced during periodontitis (*Liu et al., 2022*). It is noteworthy that *P. gingivalis* is resistant to oxidative burst killing due to its antioxidant enzymes, such as thiol, and rubrerythrin. Furthermore, these bacteria accumulate a hemin layer on the cell surface that protects the bacteria from oxidative stress (*Wang et al., 2014*; *Henry et al., 2012*).

On the other hand, IL-10, an anti-inflammatory cytokine that suppresses the inflammatory responses (*Al-Rasheeda et al., 2004*), also protects from tissue destruction by inhibiting both matrix metalloproteinases (MMPs) and receptor activators for nuclear factor-kB (RANK) systems, leading to the differentiation and activation of osteoclasts (*Garlet et al., 2006*). Stimulation with bacteria or bacterial components like LPS induce the production of inflammatory cytokines, such as interleukin 1, −6, −8, and nitric oxide (NO), in human monocytes, endothelial cells, macrophages, and gingival fibroblasts (*Gutiérrez-Venegas et al., 2005*; *Staudte et al., 2010*; *Gölz et al., 2014*). *P. gingivalis* triggers the production of NO by activating the expression of inducible nitric oxide synthases (*Sun et al., 2010*; *Brennan, Thomas & Langdon, 2003*). It is noteworthy that it can resist NO stress and maintain nontoxic intracellular NO concentrations (*Zumft, 2002*). Thus, a high concentration of NO fails to eliminate this bacterium, yet it can exert a deleterious effect on the periodontal tissue, favoring vasodilation and diminishing platelet aggregation, which contributes to gingival bleeding. These toxic effects on the surrounding tissue increase the severity of periodontitis (*Boutrin et al., 2012*). It has been suggested that the inducible nitric oxide syntase (iNOS) may be involved in periodontal pathogenesis (*Batista et al., 2002*), since common periodontal pathogens can induce the expression of iNOS in various host cells, including HGFs (*Sosroseno, Bird & Seymour, 2009*).

Additionally, cytokines and chemokines expressed by gingival fibroblasts in response to *P. gingivalis* can accumulate and their subsequent action on leukocytes is modulated due to the enzymatic activity of *P. gingivalis*-derived proteinases, that cleave and inhibit their biological properties (*Calkins et al., 1998*; *Kobayashi-Sakamoto, Isogai & Hirose, 2003*; *Palm, Khalaf & Bengtsson, 2015*). The production of *P. gingivalis* cysteine proteinases are associated with the growth and establishment of *P. gingivalis*, they are divided into arginine-specific (Rgp) and lysine-specific (Kgp) proteinases. Additionally, these cysteine proteases exert potent immunomodulatory effects on human gingival fibroblasts. The main causative factor of tissue damage involved in the disease progression, could be the gingipains of the bacterium ,even though *P. gingivalis* is considered an opportunistic pathogen. Thus, control of proteolytic enzymes of *P. gingivalis* could represent an interesting target for the treatment of periodontitis (*Torbjörn, Atika & Khalaf, 2015*).

Antimicrobial peptides (AMPs) are part of the innate defense system in the oral cavity, where cystatins play an important role. Cystatin C belongs to the type 2 family of the cystatin superfamily, it is ubiquitously distributed in plants and animals (*Shamsi & Bano, 2017*). In the parotid gland of humans, it is present in saliva at a concentration of 0.9 μg/mL (*Gorr, 2012*). The main function of cystatin C is the inhibition of cysteine proteases by binding to their active sites (*Palm, Khalaf & Bengtsson, 2015*). It also exerts several immunomodulatory functions and possesses the ability to regulate innate immune responses (*Vray, Hartmann & Hoebeke, 2002*).

The aim of this study was to assess the effect that cystatin C exerts on cytokine production, NO and ROS production by human gingival fibroblasts incubated with *P. gingivalis* in order to be able to evaluate its potential therapeutic use against one of the main etiological agent causing periodontitis, as well as its potential impact on the severity of periodontal disease.

## MATERIALS & METHODS

### Cells culture

Human gingival fibroblasts (HGFs) (ATCC, CRL-2104) were seeded at a density of $5 \times 10^3$ cells per $cm^2$ and cultured in 75 $cm^2$ culture flasks in a water saturated atmosphere at 37 °C and 5% $CO_2$ and maintained in Dulbecco's modified Eagle high glucose medium (Sigma Aldrich, Saint Louis, MO, USA), supplemented with 10% fetal bovine serum (GIBCO BRL, Gaithersburg, MD, USA), containing 10 U penicillin/25 μg streptomycin/mL (Sigma Aldrich). The fibroblasts were cultured to confluence, at a density of $2.5 \times 10^5$ cells/mL, washed twice with phosphate-buffered saline, and dissociated with 0.25% trypsin and 1 mM EDTA for 5 min at 37 °C, 5% $CO_2$ (Sigma Aldrich, Saint Louis, MO, USA). The cells were used at passages 3–7.

### Bacterial growth

*P. gingivalis* strain ATCC 33277 was cultured in brain-heart-infusion and in broth-heart-brain extract (BHI; BD Bioxon, Milan, Italy) containing 5 μg/mL of hemin (Sigma-Aldrich, Munich, Germany) and 1 μg/mL of menadione (Sigma-Aldrich) under anaerobiosis using the anaerobic BBL-GasPak jar system (BD Biosciences) at 37 °C for 24 h.

After 24 h of culturing, bacteria were harvested by centrifugation for 10 min at 10,000 rpm and then washed and resuspended in Krebs-Ringer-Glucose (KRG) buffer (120 mM NaCl, 4.9 mM KCl, 1.2 mM $MgSO_4$, 1.7 mM $KH_2PO_4$, 8.3 mM $Na_2HPO_4$, 10 mM glucose, and 1.1 mM $CaCl_2$, pH 7.3). Bacterial growth was monitored spectrophotometrically (Jenway Genova R0027, Fischer Scientific, USA) at 675 nm. The bacterial density was visually adjusted to a turbidity of 0.5 McFarland ($1 \times 10^8$ colony-forming units; (CFU/mL) (*Mc Farland, 1907*; *Emani, Gunjiganur & Mehta, 2014*). Ethical approval was given by the Ethics Committee of the School of Medicine (UNAM) with reference number C54-11.

### Antibacterial assay

Lyophilized Cystatin C was obtained from *Pichia Pastoris* (Sigma Aldrich, St. Louis, MO) and reconstituted in Tris Base NaCl Buffer (pH 7.4). Minimum inhibitory concentrations (MIC) of Cystatin C was determined using the microdilution method in 96-well microtiter

plates (Costar, Corning Life Sciences) (*Eloff, 1998*; *Jadaun et al., 2007*). Briefly, an inoculum of *P. gingivalis* ($1 \times 10^6$ CFU/ mL) containing KRG Buffer was placed in each well. Subsequently, different cystatin C concentrations (0.1, 0.3, 0.5, 0.7, 0.9 μg/mL) were incubated with the bacteria, for 1, 12, 24, and 48 h, under anaerobiosis conditions, at 37 °C. After the incubation period, 20 μL of Presto Blue Cell Viability Reagent (Invitrogen, Thermo Fisher Scientific, Waltham, MA, USA) per well were added. The plates were incubated for 30 min at 37 °C in the dark. Finally, the plates were read in a microplate reader (Multiskan SkyHigh Microplate Spectrophotometer), at a 675 nm wavelength.

## Cell viability assay

HGFs were seeded at a density of $1 \times 10^5$ cells/well in 24-well plates for 24 h, at 37 °C with 5% CO2. Different concentrations of Cystatin C (0.1, 0.3, 0.5, 0.7, 0.9 μg/mL) were added and incubated for 24 h. After incubation time, 25 μl of XTT/PBS solution (4 mg/4 mL) were added per well, for 40 min at room temperature, in the dark. Subsequently, microplate plates were read at a wavelength of 450 nm in a microplate spectrophotometer (Multiskan SkyHigh Microplate Spectrophotometer).

## Treatment of human gingival fibroblasts (HGFs) with *P. gingivalis*

Human gingival fibroblasts, at a seeding density of $5 \times 10^5$/well, were cultured in a Costar® 24-well plate (Corning Life Sciences, Corning, NY, USA) in D-MEM medium at 37 °C in an atmosphere of 5% $CO_2$. After the incubation period, fresh medium without antibiotics was added to HGFs, before they were treated with *P. gingivalis*. HGFs were stimulated with bacteria, at multiplicities of infection (MOI) of 1:100 for 24 h, and with cystatin C at a concentration of 0.3 μg/mL at 37 °C for 24 h, to perform cytokine assays, and evaluate ROS, and NO. Control groups include HGFs without stimulation or stimulated with LPS and peptidoglycans.

## Cytokine assays

For cytokine assays, HGFs were incubated with *P. gingivalis* (MOI 1:100) and /or cystatin C at a concentration of 0.3 μg/mL at 37 °C for 24 h. Control groups included HGFs without stimulation or stimulated with LPS 100 ng/mL (LPS from *Escherichia coli* O111:B4; Sigma Aldrich), or with peptidoglycan 10μg/mL (Peptidoglycan from *Staphylococcus aureus*; Sigma Aldrich). ELISAs were performed to determine TNF-α, IL-1 β, and IL-10, using the Ready-Set-Go! ELISA kits (Cytokine ELISA Protocol; BD Biosciences, San Diego, CA, USA), according to the manufacturer's protocol. Dilutions were prepared in dilution buffer. Briefly, 96-well flat-bottom plates (Costar®, Corning Life Sciences) were coated with anti-human TNF-α, IL-1 β, or IL-10 monoclonal antibodies (BD Biosciences, Pharmingen). After blocking with the assay solution (PBS−0.5% casein diluted in 1 M NaOH) overnight at 4 °C to avoid non-specific binding, 100 μL of standard TNF-α, IL-1 β, or IL-10 (BD Bioscience, Pharmingen) of supernatants were added. The microplate was washed to remove unbound enzyme-labeled antibodies. The amount of horseradish peroxidase bound to each well was determined by the addition of a substrate solution. The reaction was stopped by the addition of sulfuric acid and the plates were read at 405 nm (ELISA microplate reader; Bio-Rad, Hercules, CA, USA).

The cytokine concentration was calculated by regression analysis from a standard curve. The detection limit of the assay was 15 to 2000 pg/mL.

## Measurement of NO production

The NO production by HGFs incubated with *P. gingivalis* and/or cystatin C at 37 °C was assayed by measuring the accumulation of nitrate in culture supernatants. Briefly, HGFs were stimulated with *P. gingivalis* (MOI 1:100) and with 0.3 μg of cystatin C, at 37 °C for 24 h. Thereafter, 100 μL of Griess reagent (1% sulphanilamide, 0.1% naphthylethylene diamine dihydrochloride, and 2.5% phosphoric acid) (Sigma Aldrich) were added at equal volumes of culture supernatants in a 96- well plate (Costar®; Corning Life Sciences) and left at room temperature for 30 min. The absorbance of these supernatants was read at 550 nm (Multiskan SkyHigh Microplate Spectrophotometer) and the nitrate concentrations were calculated from a standard curve established with serial dilutions of $NaNO_2$ (Sigma-Aldrich) in the culture medium. Control groups included HGFs without stimulation or stimulated with LPS or peptidoglycan.

## Detection of Reactive Oxygen Species (ROS)

HGFs were seeded on 24-well plates (Costar®; Corning Life Sciences) at a density of ($5 \times 10^5$), infected with *P. gingivalis* (MOI 1:100) and stimulated with 0.3 μg/mL of cystatin C at 37 °C for 24 h. The cells were incubated with 100 μg/ mL of 2,7 dichlorodidrofluoroescein diacetate (H2-DCFDA) [2 μM/mL] for 30 min in the dark at room temperature. Cells were rinsed twice with PBS, pH 7.2 and detached from the wells with 0.25% Trypsin/EDTA (Sigma Aldrich). The samples were resuspended in PBS, pH 7.2, with 1% FBS and analyzed on a FACS Canto II BD Biosciences flow cytometer. Data analysis was performed using FlowJo software (USA). Control groups included HGFs without stimulation or stimulated with LPS or peptidoglycan.

## Statistical analysis

Experimental and control conditions were statistically compared for significance using analysis of variance (ANOVA), followed by Benferroni correction. The predetermined level of significance was $p < 0.05$. Statistical analysis was performed with the GraphPad, Prism v.6 software (GraphPad Software, Inc., CA, USA).

## RESULTS

### Effects of cystatin C on growth of *P. gingivalis* and viability of HGFs

The antimicrobial activity of cystatin C on *P.gingivalis* was analyzed in a time and *dose-dependent manner* as shown in (Fig. 1A). It reached its maximal antimicrobial activity *at 24 h with concentrations* between 0.1 and 0.3 μg/mL.

The concentration of 0.3 μg/mL inhibited 75% of bacteria growth after 24 h of incubation when compared to the control group ($p < 0.05$). Inhibition of bacterial growth (83.3%) was observed after 48 h of culture ($p < 0.05$). At a concentration of 0.9 μg/mL a marked growth inhibition was observed throughout the incubation time. All the analyzed concentrations of cystatin C showed no effect on the viability of HGFs cell, as illustrated in (Fig. 1B).

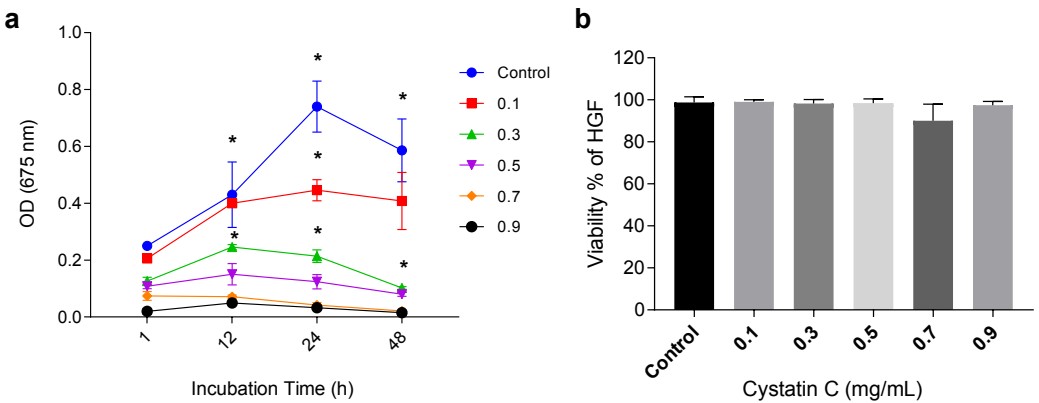

**Figure 1** **Effect of cystatin C on *Porphyromonas gingivalis* growth and cell viability.** Effect of cystatin C on (A) *Porphyromonas gingivalis* growth and (B) Cell viability of HGFs. The results are expressed as mean ± SD of the average of five independent assays. Statistical differences are expressed as (*) $p < 0.05$ when compared to non-treated control *P. gingivalis* bacteria.

These findings reveal the antimicrobial activity of cystatin C against *P. gingivalis* and did not affect the viability of HGFs. Hence, we decided to perform all the experimental assays with a cystatin C MIC at 0.3 µg/mL.

## Effect of cystatin C on the production of pro- and anti-inflammatory cytokines

TNF-$\alpha$ and IL-1 $\beta$ were evaluated in supernatants of HGFs incubated with *P. gingivalis* and cystatin C (0.3 µg/mL) for 24 h. *P. gingivalis* induced the production of 1000 pg/ mL and 750 pg/mL of TNF-$\alpha$ and IL-1 $\beta$, respectively, when compared to the control group ($p = 0.0001$) (Figs. 2A and 2B). However, when HGFs were incubated with the bacteria and cystatin C, a statistically significant decrease was observed in the TNF-$\alpha$ ( $p = 0.0001$) and IL-1 $\beta$ ($p < 0.05$) productions, compared to HGFs. In contrast, no changes were observed in IL-10 production by HGFs incubated with *P. gingivalis* alone, when compared to controls, whereas cystatin C stimulated de production and secretion of IL-10 (500 pg/mL). Furthermore, the co-incubation of *P. gingivalis* with cystatin C significantly increased the production of IL-10 (900 pg/mL), when compared with the control group and with HGFs infected with the bacterium ($p = 0.0001$), (Fig. 2C). These results suggest that cystatin C participates in the regulatory inflammatory process, by reducing inflammatory cytokines and increasing anti-inflammatory cytokines.

## Cystatin C decreases ROS and NO production on HGFs incubated with *P. gingivalis*

A significant increase was observed in the production of ROS and NO in HGFs incubated with *P. gingivalis*, compared to the controls ($p = 0.0001$). No significant differences were observed in the production of ROS in HGFs incubated with cystatin C ($p > 0.05$) (Fig. 3A). In contrast, a significant decrease in ROS was observed after the incubation of HGFs with *P. gingivalis* and cystatin C, compared to the control ($p = 0.001$), (Fig. 3A).

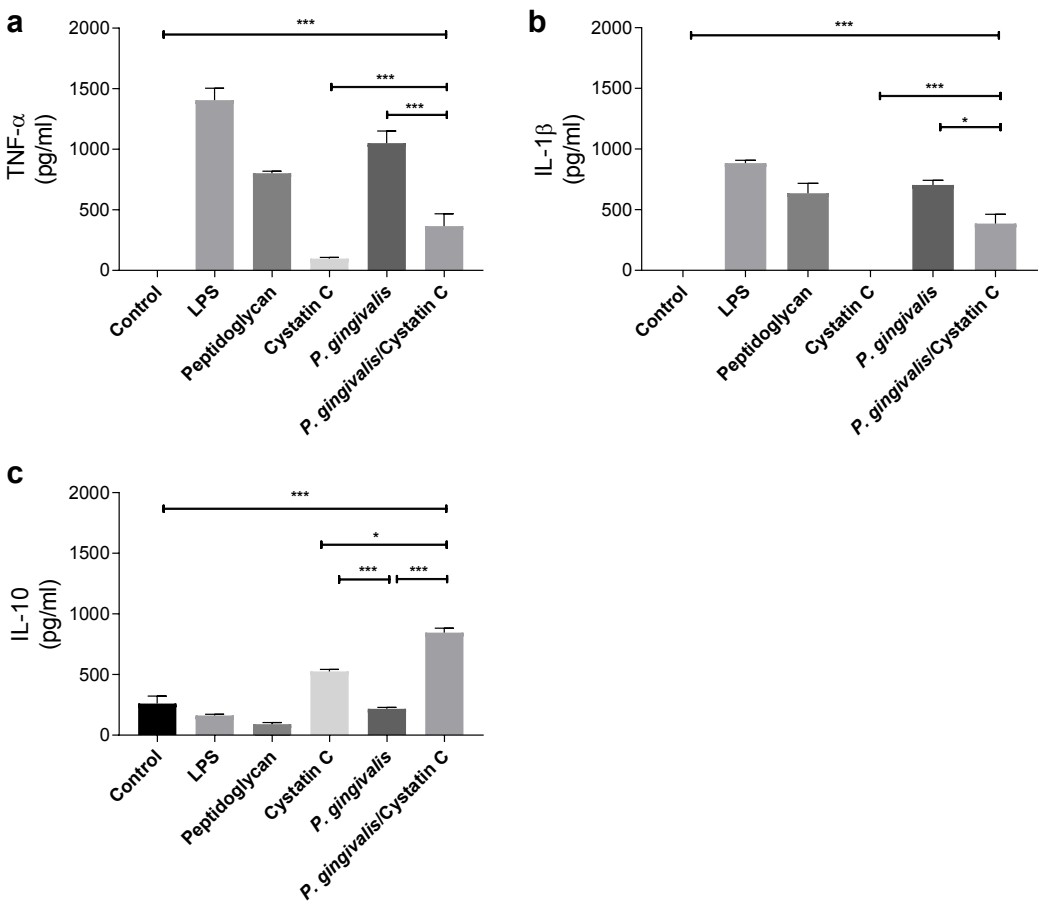

**Figure 2** **Cytokines assays in HGFs incubated with *P. gingivalis* and stimulated with cystatin C.** (A) Expression of TNF-$\alpha$, (B) IL-$\beta$, (C) IL-10. The results are expressed as mean $\pm$ SD of the average of five independent assays. Statistical differences are expressed as (*) $p < 0.05$, (**) $p < 0.001$, (***) $p < 0.0001$, compared to control groups.

Furthermore, a significant increase of NO (9 $\mu$M) was observed after the incubation of HGFs with *P. gingivalis,* when compared with the control group ($p = 0.0001$). Yet when HGFs were incubated with *P. gingivalis* and cystatin C, a decrease of NO (3 $\mu$M) ($p = 0.001$) was observed with regard to the incubation with *P. gingivalis* alone (Fig. 3B).

# DISCUSSION

In this study, we analyzed the antimicrobial activity of cystatin C against *P gingivalis*, which contributes to the development of chronic periodontitis. The immunological responses occurring in HGFs after the infection with this key periodontal pathogen were evaluated. *P. gingivalis* exhibits a variety of virulence factors that enable it to colonize oral soft tissues and evade immune responses. It has been demonstrated that *P. gingivalis* triggers and suppresses the immune responses in HGFs, suggesting that the pathogenic effects of *P. gingivalis* are mainly related to the action of gingipains, which participate in the inflammatory and immune response of HGFs (*Palm, Khalaf & Bengtsson, 2015*; *Bengtsson,*

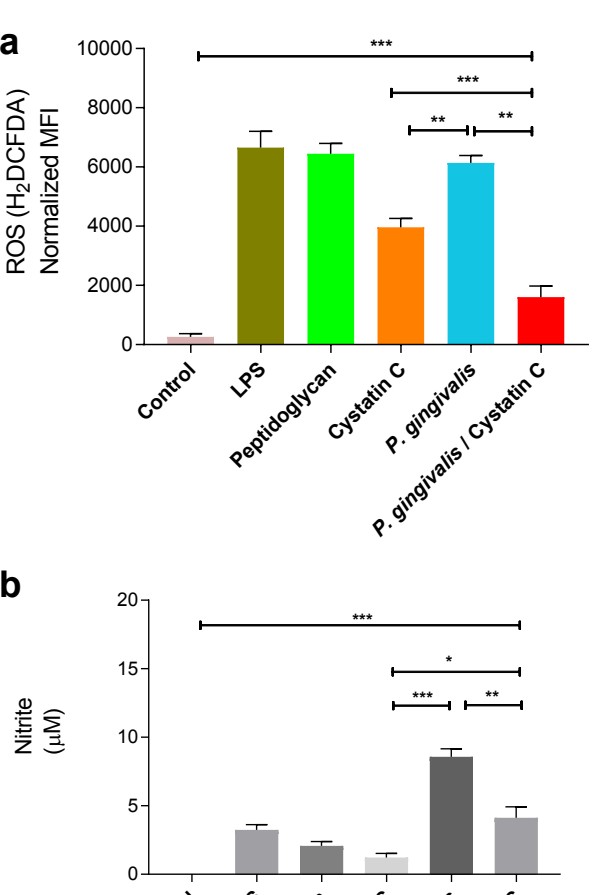
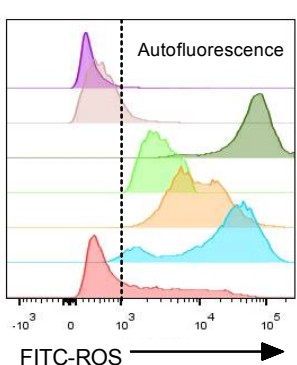

**Figure 3** **ROS and NO production *in Porphyromonas gingivalis* incubated with HGFs and cystatin C.**
*Porphyromonas gingivalis* incubated with HGFs and cystatin C. (A) ROS production in HGFs infected with
*P. gingivalis*. (B) Expression of nitric oxide in HGFs incubated with *P. gingivalis*. The results are expressed
as mean ± SD of the average of five independent assays. Statistical differences are expressed as (*) $p <$
0.05, (**) $p < 0.001$, (***) $p < 0.0001$, compared to control groups.

*Khalaf & Khalaf, 2015*). Additionally, *P. gingivalis* has a direct modulatory function on
the immune response of fibroblasts through the catalytic activities of gingipains, targeting
fibroblast-derived inflammatory mediators at the protein level (*Palm, Khalaf & Bengtsson,
2013*). *P. gingivalis* secretes three related cysteine proteases (gingipains), which constitute its
main virulence factors. Two gingipains are specific for Arg-Xaa peptide bonds (HRgpA and
RgpB), whereas Kgp cleaves after a Lys residue (*Imamura, 2003*). Interestingly, gingipains
are involved in the disruption of host defense inflammatory reactions and hinder *P.
gingivalis* clearance by the immune system (*Uehara et al., 2008*; *Guo, Nguyen & Potempa,
2010*). Human gingival fibroblasts play an important part in the innate immune system by
sensing microbial invasion and responding to it by producing and secreting inflammatory

mediators. HGFs recognize *P. gingivalis* during the early stages of periodontitis and establish an inflammatory response in the periodontal tissue (*Palm, Khalaf & Bengtsson, 2015*). The secretion of TNF-$\alpha$ and IL-1 $\beta$ by HGFs favor the recruitment of macrophages and neutrophils to the site of infection, as well as the expression of MMP-1, MMP-13, MMP-8, and MMP-9, which contribute to the degradation of the extracellular matrix of the periodontal tissue as well as the reabsorption of bone tissue (*Ara et al., 2009*; *Song et al., 2021*; *Cheng et al., 2020*; *Franco et al., 2017*).

Interleukin-1 $\beta$ (IL-1 $\beta$), belongs to the IL-1 family and plays an important role against microbial infections and participates regulating innate immune and inflammatory responses. The upregulation of IL-1 $\beta$ during P. *gingivalis* infection suggests that IL-1 $\beta$ is a critical cytokine in the host's defense against *P. gingivalis* infection during the initial phases of inflammation (*Dinarello, 2009*). In the early stages of *P. gingivalis* infection, IL-1 $\beta$ plays an important role in combating the invading pathogen as part of the innate immune response and participates in almost all events involved in the activation and regulation of inflammation (*Menu & Vince, 2011*). This kind of inflammasome-independent IL-1 $\beta$ activation can substantially contribute to tissue inflammation (*Latz, Xiao & Stutz, 2013*).

We now demonstrate that cystatin C down-regulates the production of IL-1 $\beta$ and TNF-$\alpha$ in HFGs co-incubated with *P. gingivalis*. Our finding is in accordance with the literature, where cystatin C has been shown to down-regulate the production of IL-1 $\beta$ and TNF-$\alpha$ in monocytes stimulated with bacterial LPS (*Gren et al., 2016*). In addition to cystatin C, other salivary antimicrobial peptides, such as histatin 5 and histatin 1, also down-regulate inflammatory cytokines like IL-6, IL-8, IL-1 $\beta$, and TNF-$\alpha$ in fibroblasts and macrophages (*Imatani et al., 2000*; *Lee et al., 2021*).

Our data also show that cystatin C enhances IL-10 production by HFGs incubated with *P. gingivalis*, which could represent an important mechanism to inhibit an excessive inflammatory response of HGFs to the *P. gingivalis* infections. The cytokine IL-10 can inhibit pro-inflammatory responses, due to its ability to reduce the production of TNF-$\alpha$, IL-6, and IL-1 cytokines (*Sun et al., 2020*). Our results suggest that cystatin C could be an important multifunctional modulator of the innate immune responses in HGFs.

In addition to cytokine production, HGFs also produce microbicidal mediators such as ROS and NO, when they are infected with *P. gingivalis*. High doses of these molecules have been shown to be cytotoxic to periodontal tissue (*Nogueira et al., 2016*), since their excessive production may lead to tissue breakdown, including inhibition of energy-generating enzymes, triggering DNA injury, oxidation and nitration reactions (*Wang, Huang & He, 2019*; *Bodis & Haregewoin, 1993*). ROS causes oxidative damage to proteins and DNA, it interferes with cell growth, and induces apoptosis in gingival fibroblasts, causing periodontitis (*Kanzaki et al., 2017*; *Cheng et al., 2015*; *Tomofuji et al., 2006*; *Maruyama et al., 2011*). In addition to the damage caused by ROS, an increase of iNOS expression and NO concentration also leads to severe damage related to bone resorption, as shown in an experimental rat model of periodontitis (*Wang, Huang & He, 2019*). Thus, many inflammatory mediators are crucial for the development of early periodontal disease, where NO is one of the main inflammatory factors (*Pacher, Beckman & Liaudet, 2007*)

Our data now demonstrated that *P. gingivalis* stimulates NO release by HGFs and that the co-incubation of the bacterium with cystatin C significantly down-regulates both ROS and NO productions. These findings are in accordance with the literature, showing that other peptides, such as hBD3 and sublancin, also reduce the production of ROS in endothelial cells and NO in peritoneal macrophages, respectively (*Wang, Huang & He, 2019*; *Bian et al., 2017*). The results of our study suggest that NO expression could lead to the gradual progression of periodontitis after proinflammatory cytokine production by HGFs infected by *P. gingivalis* and that cystatin C protects from tissue damage through the reduction of these free radicals. The importance of ROS in periodontal diseases was previously demonstrated by *Cheng et al. (2015)*, who showed that LPS from *P. gingivalis* up-regulated ROS in periodontal ligament fibroblasts (*Cheng et al., 2015*; *Gölz et al., 2014*). The release of inflammatory mediators including interleukins, chemokines, adhesion molecules, and ROS could be could be triggered by bacteria LPS (*Goraca et al., 2013*; *Melo et al., 2010*; *Sanikidze et al., 2006*; *Bykov et al., 2003*).

Antimicrobial peptides are included in the immune innate defense system in the oral cavity (*Greer, Zenobia & Darveau, 2013*). The antimicrobial peptide cystatin C belongs to the type 2 family of the cystatin superfamily, it is ubiquitously distributed in plants, animals, and microorganisms (*Shamsi & Bano, 2017*). Saliva from the parotid gland of humans contains 0.9 µg/mL of Cystatin C (*Gorr, 2009*). The main function of cystatin C is the inhibition of cysteine proteases, by binding to their active sites, evading the cleavage of peptide bonds (*Van Wyk et al., 2016*). The mechanisms leading to the reduction of the inflammatory mediators by cystatin C are possibly explained by observations made with a homologous molecule, DsCistatin, isolated from the tick *Dermacentor silvarum*. This peptide was shown to be internalized by endocytosis in mouse macrophages stimulated with LPS from *Borrelia burgdorferi*. It reduced the inflammatory cytokines IL-1 $\beta$, IFN-$\gamma$, TNF-$\alpha$, and IL-6 by the degradation of the TRAF6 protein, thereby preventing the phosphorylation of I$\kappa$B$\alpha$ and the subsequent nuclear transport of NF-$\kappa$B, leading to the decrease of inflammatory cytokines (*Sun et al., 2018*). We speculate that cystatin C possibly follows this route to reduce inflammatory mediators in HGFs incubated with *P. gingivalis*.

Our data now show that cystatin C possibly plays an important antimicrobial and anti-inflammatory role that regulates the response of human gingival fibroblast towards *P. gingivalis*, helping to avoid tissue damage and destruction.

## CONCLUSIONS

Cystatin C exhibits a dual activity during *P. gingivalis* infection. Antimicrobial activity was demonstrated without cytotoxic effects on HGFs. Furthermore, Cystatin C also exhibited immunomodulatory functions, decreasing the inflammatory response of fibroblasts. Knowledge on the immunomodulatory properties of cystatin C could aid in the design of new therapeutic approaches to improve the treatment of periodontal diseases.

## ACKNOWLEDGEMENTS

We thank Drs Daniela Cortés Hernández and Dulce Verónica Rivero Gamallo for their assistance in the culture of bacteria and human fibroblasts during the initial phase of the study and Rocely Cervantes Sarabia for her assistance in cytotoxicity assays. The authors thank to the Posgrado de Ciencias Biológicas, Universidad Nacional Autónoma de México.

### Funding

This work was supported by grant #IN218419 from PAPITT, DGAPA, UNAM, Mexico City, and by Universidad Anahuac México Campus Norte. Blanca Esther Blancas-Luciano is supported by CONACYT grant #424031 for her doctoral studies. The funders had no role in study design, data collection and analysis, decision to publish, or preparation of the manuscript.

### Grant Disclosures

The following grant information was disclosed by the authors:
PAPITT, DGAPA, UNAM, Mexico City: #IN218419.
Universidad Anahuac México Campus Norte.
CONACYT: #424031.

### Competing Interests

The authors declare there are no competing interests.

### Author Contributions

- Blanca Esther Blancas-Luciano conceived and designed the experiments, performed the experiments, analyzed the data, prepared figures and/or tables, authored or reviewed drafts of the article, and approved the final draft.
- Ingeborg Becker-Fauser conceived and designed the experiments, analyzed the data, authored or reviewed drafts of the article, and approved the final draft.
- Jaime Zamora-Chimal conceived and designed the experiments, performed the experiments, authored or reviewed drafts of the article, and approved the final draft.
- José Delgado-Domínguez conceived and designed the experiments, performed the experiments, authored or reviewed drafts of the article, and approved the final draft.
- Adriana Ruíz-Remigio conceived and designed the experiments, performed the experiments, authored or reviewed drafts of the article, and approved the final draft.
- Elba Rosa Leyva-Huerta conceived and designed the experiments, analyzed the data, authored or reviewed drafts of the article, and approved the final draft.
- Javier Portilla-Robertson conceived and designed the experiments, analyzed the data, authored or reviewed drafts of the article, and approved the final draft.
- Ana María Fernández-Presas conceived and designed the experiments, performed the experiments, analyzed the data, authored or reviewed drafts of the article, and approved the final draft.

## Data Availability

The raw data is available in the Supplemental Files.

## Supplemental Information

Supplemental information for this article can be found online at http://dx.doi.org/10.7717/peerj.14232#supplemental-information.

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
