# Peer review of "Antimicrobial and anti-inflammatory activity of Cystatin C on human gingival fibroblast incubated with Porphyromonas gingivalis"

_PeerJ, doi:10.7717/peerj.14232_

## Round 0.1 · original submission · Major Revisions

Dear authors:

Thank you for submitting this manuscript that both reviewers and myself think there are several interesting results and worth looking into further. I kindly request that authors check their manuscript carefully, perhaps the use of an editing service to correct the overall writing and check all aspects of the manuscript. Also, I kindly request that authors revise carefully the comments done by the two expert reviewers and respond carefully to their concerns. I specifically ask to carefully revise the MIC determination which reviewer 2 pointed out.

Your manuscript once submitted will be sent again to the two experts to assess the corrections.

Thank you so much for choosing Peer J and I am looking forward to your revised manuscript.

Best regards,
Bernardo

·

Basic reporting

This manuscript represents a simple but well done study demonstrating the antimicrobial peptide cystatin C over the periodontal pathogen P. gingivalis, confirming its antibacterial activity and demonstrating it also counteracts the inflammatory effect produced by the bacteria in fibroblasts, due the release of cytokines and the production of ROS and NOS, hence suggesting a broad mechanism of action of this peptide counteracting periodontal disease.
I have no major concerns but a list of minor concerns related to writing, since although the paper is written in proper English and is readable it contains several mistakes (in fact it seems a preliminary draft rather than a finished manuscript).
Minor concerns
1) Abstract :“the effect of Cystatin C effect on” remove “effect on”
2) Abstract : use italics for “Porphyromonas gingivalis”
3) L 55 “periodontopathogen” change to: periodontal pathogen
4) L 61 “Baek et al., 2013)” add “.”
5) L 63 “( St” extra space
6) L 65 “inflammatory such as mediators nitric oxide cytokines,and reactive oxygen species (ROS), and nitric oxide” rephrase.
7) L 72 “severity. and” remove “.”
8) L 88 “pathogen killing and cytotoxic effects on periodontal tissues and pathogen killing” “pathogen killing” is repeated.
9) L 92 “antioxidant enzymes, such as thiol,” thiols are not enzymes but functional groups, do you mean thiol containing enzymes or other molecules containing thiols?
10) L 118 “they .are” remove “.”
11) L 131 “2002)” add “.”
12) L 134 “agent” change to “agents”
13) L 142 “at” remove italics.
14) L 155 “24 h” add “.”
15) L 208 “of sulfuric acid” which concentration?
16) L 340 “(Pacher Beckman, & Liaudet, 2007” add “).”
17) L 352 “could be.could be” and change “bacteria” to “bacterial”
18) L 352 “Bykov et al2003)” lack of space and lack of “.”

Experimental design

it is well done, I have no concerns.

Validity of the findings

They are valid, I have no concerns.

Additional comments

please adress my concerns and carefully revise you manuscript to correct other possible errors I may overlook.

·

Basic reporting

To change the surname of one of the author due first has been written such Domónguez and in page 6 such Domínguez.
Is mandatory the change of orange color to black in all document.
Title makes emphasis in antimicrobial activity of Cystatin C, however, in the aim of the paper authors did not mention it.
Line 46 dot underlined
line 65 after coma space and then and "65 inflammatory such as mediators nitric oxide cytokines,and reactive oxygen species (ROS), and"
Line 68 before "Even though" a dot.
Line 72 after severity is a dot, must be changed by a coma.
Line 83 Before These pro-inflammatory and after dot a space must be given.
Line 90 "ROS production in mitochondria (mtROS) was significantly enhanced" was must be changed by were due acronym is Reactive oxygen species. In the same line after enhanced there are two comas.
Line 98 MMP was previously described in line 77 then is not necesary define it again in same line is defined RANK acronym however, it was cited before without definition. Please check all acronyms before.
Line 118 "118 establishment of P. gingivalis, they .are divided into" between they and are there is a dot, please remove it.
Line 119 "119 (Kgp) proteinases .Additionally" TO remove the space between proteinases and the dot, besides after dot should have an space between dot and Additionally.
Line 131 To write a dot after parenthesis.
Line 141 water and 142 satured must be stay continuous, without enter.
Line 142 at is ins italic, please change to normal.
Line 155 after jaar system there is a dot then parenthesis and finally 24 h without the respective dot at the end.
Line 179 CO2, the two in subscript
Line 180. 25 µl of XTT/PBS solution (4 mg/4ml) were added per well. This meant that is equal to 1 mg/mL? or 4 mg/mL?
Line 186 to remove |
Line 187 To change D-MEM by DMEM
Line 227 to remove the space.
Line 229 and 230 after 2.7 remove the enter.
Line 230. Concentration should go before reagent.
Line 260. 1000 pg/ mL, after slash there is a space and then mL, please remove the space.
Line 281. To change the color
Line 352. Could be is repeated.

Experimental design

Determination of minimal inhibitory concentration must be done visually not with the use of spectrophotometers. In this point i´d like to know wich criteria did authors used to define susceptible wich value obtained with spectrophotometer, which is the diference with minimal bactericidal concentration?
Line 190 Why cystatin at a final concentration of 0.3 µg/mL was chosen?
Line 192 Which concentration of LPS or peptidoglycan?
Line 223 and 224 Could author say which concentration of LPS or peptidoglycan in this experiment.

Validity of the findings

Author must give more information in discussion, at the point where they suggest that NO expression could lead to the gradual progression of periodontitis after proinflammatory cytokine production by HGFs infected by P. gingivalis. This must be supported with more literature and should be referred.
If it is knowns that saliva contains 0.9 µg/mL of Cystatin C why authors decided to use 0.3? Is this not a subMIC effect? In physiolocally contidions is overrated the the concentration, what if author considerer increase the concentration, I mean what if they test 0.9 (mormal concentration) + 0.3 according they saw. Or could you discuss what happens in periodontal disease according with cystatin C? It dreceases? stay equal?

Additional comments

Current work is interesting, authors focus in one of the most prevalent oral diseases, However, work has several syntaxis errors rason why is mandatory a conscientious review. Several spelling erros, the use of differents colors. As said before a mandatory review.

---

## Round 0.2 · Minor Revisions

Dear authors,

Based on the comments made by reviewer 2, I decided to also review the manuscript. Please check the complete manuscript once again but to save you some time, here are the most critical issues in the manuscript that need to be revised. Once this issues are fixed, the manuscript will be suitable for publication. Please provide a comment regarding the MIC and read my last comment on Figure 1.

One of the reviewers has pointed out that there are some grammatical issues with your manuscript, here I provide specifically those that are critical:
In line 66 a space is missing with the reference list.
In line 72 please correct the punctuation at the end of the line, a semi-colon is advice here.
In line 79, please change resorption to reabsorption.
Line 96, please remove “On the other hand” is not needed here.
Line 118, there is an additional . here, please remove
Line 121, please correct the coma in “,even though”
In line 156 please correct to 10,000.
Please correct CO2 in line 180
Please correct in line 246 the space missing in P. gingivalis.
Please revise reference in line 313.
Please in line 795 complete the availability of the data presented here.
Regarding Figure 1, please indicate the statistical significance on each dataset and make the figure a bit bigger so the lines can be seen properly with the statistical significance for each data point. The way is presented here is not conventional and perhaps this has led to the comment made by Reviewer 2.

I am looking forward for your revised manuscript.
Warm regards,
Bernardo

·

Basic reporting

The manuscript is now OK

Experimental design

The manuscript is now OK

Validity of the findings

The manuscript is now OK

Additional comments

Thanks for adressing my comments and improving the writing of your manuscipt.

·

Basic reporting

Authors reviwed the document, however, this was done partiatily since there are, still, several grammar erros,
Annotations were directly in word document, is mandatory a proper review of this paper again.
Some comments and observations were written in word document such as changes control.

Experimental design

There are, still, several errors due a deeply revesion was performed.
Theare still doubs about the diferences between mMIC

Validity of the findings

No comments

Additional comments

No comments

---

## Round 0.3 · accepted · Accept

Dear authors,

After revising the current version of this manuscript, I thank you for addressing all the comments. I think this study is relevant and well written and designed. I thank you for submitting this manuscript to PeerJ.

With warm regards,
Bernardo